# Sex Differences in Prognosis of Heart Failure Due to Ischemic and Nonischemic Cardiomyopathy

**DOI:** 10.3390/jcm12165323

**Published:** 2023-08-16

**Authors:** Antonio de Padua Mansur, Antonio Carlos Pereira-Barretto, Carlos Henrique del Carlo, Solange Desirée Avakian, Naomi Kondo Nakagawa, Luiz Antonio Machado Cesar, Edimar Alcides Bocchi

**Affiliations:** 1Serviço de Prevencao, Cardiopatia na Mulher e Reabilitação Cardiovascular, Instituto do Coracao (InCor), Hospital das Clinicas HCFMUSP, Faculdade de Medicina, Universidade de Sao Paulo, Sao Paulo 05403-900, SP, Brazil; pereira.barretto@incor.usp.br; 2Hospital Dia, Instituto do Coracao (InCor), Hospital das Clinicas HCFMUSP, Faculdade de Medicina, Universidade de Sao Paulo, Sao Paulo 05403-900, SP, Brazil; del_carlo@uol.com.br; 3Unidade Clinica de Valvopatias, Instituto do Coracao (InCor), Hospital das Clinicas HCFMUSP, Faculdade de Medicina, Universidade de Sao Paulo, Sao Paulo 05403-900, SP, Brazil; solange.avakian@incor.usp.br; 4Departamento de Fisioterapia, Faculdade de Medicina, Universidade de Sao Paulo, Sao Paulo 01246-903, SP, Brazil; naomi.kondo@fm.usp.br; 5Unidade Clinica de Coronariopatias Crônicas, Instituto do Coracao (InCor), Hospital das Clinicas HCFMUSP, Faculdade de Medicina, Universidade de Sao Paulo, Sao Paulo 05403-900, SP, Brazil; dcllucesar@incor.usp.br; 6Unidade Clinica de Insuficiencia Cardiaca, Instituto do Coracao (InCor), Hospital das Clinicas HCFMUSP, Faculdade de Medicina, Universidade de Sao Paulo, Sao Paulo 05403-900, SP, Brazil; edimar.bocchi@incor.usp.br

**Keywords:** heart failure, cardiomyopathy, ischemic heart disease, prognosis, women, men

## Abstract

Background: Limited research has explored sex-specific differences in death predictors of HF patients with ischemic (iCMP) and nonischemic (niCMP) cardiomyopathy. This study assessed sex differences in niCMP and iCMP prognosis. Methods: We studied 7487 patients with HF between February 2017 and September 2020. Clinical features and echocardiographic findings were collected. We used Kaplan–Meier, Cox proportional hazard models, and chi-square scores of Cox regression to determine death predictors in women and men. Results: The mean age was 64.3 ± 14.2 years, with 4417 (59%) males. Women with iCMP and niCMP exhibited a significantly higher mean age, higher mean left ventricular ejection fraction, and smaller left ventricular diastolic diameter than men. Over 2.26 years of follow-up, 325 (14.7%) women and 420 (15.7%) men, and 211 women (24.5%) and 519 men (29.8%) with niCMP (*p* = NS) and iCMP (*p* = 0.004), respectively, died. The cumulative incidence of death was higher in men with iCMP (log-rank *p* < 0.0001) but similar with niCMP. Cox regression showed chronic kidney disease, diabetes, stroke, atrial fibrillation, age, and myocardial infarction as the main predictors of death for iCMP in women and men. Conclusions: Women exhibited a better prognosis than men with iCMP, but similar for niCMP. Nevertheless, sex was not an independent predictor of death for both CMP.

## 1. Introduction

Heart failure (HF) is a complex cardiovascular condition associated with significant morbidity and mortality worldwide. It substantially burdens public health, with approximately 6 million people in the United States currently affected by HF [1]. The prevalence of HF is expected to rise due to factors such as an aging population and the increasing prevalence of risk factors, including hypertension, diabetes, and obesity. Despite advancements in HF management, the 5-year mortality rate remains high, ranging from 30% to 50% [2].

Existing studies have demonstrated that the etiology of HF influences patient outcomes, with iCMP patients exhibiting higher mortality rates than niCMP patients [3]. Interestingly, sex differences have also been observed in HF outcomes. Despite having higher hospitalization rates, women tend to exhibit better survival rates than men [4,5]. However, there is a paucity of research investigating sex-specific differences in mortality and predictors of death among individuals with HF, particularly those with ischemic (iCMP) and nonischemic (niCMP) cardiomyopathy.

Therefore, this study aims to comprehensively analyze mortality rates and identify predictors of death in women and men diagnosed with iCMP and niCMP. By examining sex-related differences in mortality and predictors of death, this study aims to provide valuable insights into the management and treatment of HF for both women and men.

## 2. Materials and Methods

This retrospective study was conducted on a cohort of 7487 patients diagnosed with chronic HF at our Heart Institute from February 2017 to September 2020, and the Research Ethics Committee approved the research project. 

The study included patients diagnosed with HF based on the Framingham criteria for HF diagnosis and echocardiographic measurements. Ischemic cardiomyopathy (iCMP) was defined as patients with a history of coronary artery disease, including known chronic angina pectoris, previous myocardial infarction (MI), coronary artery bypass grafting (CABG), or previous percutaneous coronary intervention (PCI) with or without stenting, and with more than a 70% luminal reduction in the left main coronary artery, proximal left anterior descending, or two or more coronary arteries with a significant area of myocardium depending on this artery flow. In this study, ischemic cardiomyopathy encompassed all HF phenotypes of patients with HF and significant coronary artery disease. On the other hand, nonischemic cardiomyopathy (niCMP), dilated, idiopathic, and hypertensive CMP were diagnosed in the presence of normal or nonobstructive coronary arteries [6]. Echocardiographic data were collected from patients with both baseline and the closest echocardiogram available at the end of the study. The baseline echocardiography means an echocardiographic examination within six months before the study entry.

The study’s primary outcome was cardiovascular death, encompassing fatal myocardial infarction, stroke, or other cardiovascular causes of death. Mortality data were obtained from the patient’s medical records or through the individual registration status on the Federal Revenue’s website [7].

Several clinical characteristics were analyzed, including age, the prevalence of comorbidities, the number of HF hospitalizations, and cardiac surgical interventions. The comorbidities examined were diabetes (defined as glycemia ≥ 126 mg/dL or glycated hemoglobin > 6.5% or under hypoglycemic drug), significant chronic kidney disease (CKD) (defined as creatinine ≥ 2 mg/dL), atrial fibrillation (AF), MI, and stroke. Cardiac surgical interventions assessed included PCI, CABG, valve replacement, pacemaker implantation, cardiac resynchronization therapy (CRT), implantable cardioverter defibrillators (ICD), and cardiac transplantation. Echocardiographic data included left ventricular ejection fraction (LVEF) and left ventricular diastolic diameters (LVDD) at baseline and the end of the study. HF was categorized based on LVEF as HF with a reduced ejection fraction (HFrEF) when LVEF ≤ 40%, HF with a mid-range ejection fraction (HFmrEF) when LVEF was between 41% and 49%, and HF with a preserved ejection fraction (HFpEF) when LVEF ≥ 50%. HFpEF was diagnosed in patients with symptoms (NYHA functional class II to IV) of HF requiring treatment with diuretics, structural heart disease (left atrial enlargement or left ventricular hypertrophy), and signs of diastolic dysfunction on echocardiograms. Patients hospitalized with HF or BNP > 150 pg/mL were also diagnosed with HFpEF.

### Statistical Analysis

Statistical analysis involved presenting continuous variables as the mean and standard deviation and categorical variables as frequencies and percentages. The normality of the data was assessed using the Kolmogorov–Smirnov test. Student’s *t*-test and analysis of variance were employed to compare continuous variables between groups, while the chi-square test was used for categorical variables. A two-sided probability value of <0.05 was considered statistically significant. Multiple imputation was used to impute missing baseline and follow-up LVDD values. Multiple imputations used the MCMC method to deal with missing data. The imputed datasets were analyzed separately and combined to produce a single result, considering the uncertainty caused by missing data. The cumulative incidence of all-cause death was analyzed using the Kaplan–Meier (K-M) method with Šidák multiple-comparison adjustment. Cox proportional hazards models were utilized to identify variables independently associated with all-cause death. The chi-square score of the Cox proportional hazards model was used to determine the most robust predictors of all-cause death. The dependent variable in the Cox proportional hazards model was death, while the covariates included were those with *p* < 0.1, such as age, sex, MI, diabetes, stroke, CKD, AF, baseline LVEF at echocardiogram, all coronary surgical interventions (PCI + CABG), and the implantation of all devices (pacemaker + cardiac resynchronization therapy + implantable cardioverter defibrillator). The statistical analyses were performed using the SAS^®^ Studio package (SAS Institute, Cary, NC, USA).

## 3. Results

Table 1 shows the clinical characteristics and echocardiographic data for all 7483 patients with HF and those with nonischemic and ischemic CMP studied during a mean follow-up period of 2.26 years.

In patients with HF and niCMP, men had a higher prevalence of idiopathic CMP (1991 (40.7%) vs. 1468 (30.1%); *p* = 0.017), and women had more hypertensive CMP (687 (14.1%) vs. 737 (15.1%); *p* < 0.001). Diabetes, AF, MI, CKD, and stroke in all patients, AF, diabetes, CKD, and stroke in niCMP, and diabetes, MI, AF, CKD, and stroke in iCMP were the most prevalent comorbidities. The niCMP patients had more pacemaker implantation, ICD, CRT, and transplants (*p* < 0.0001 for all). Baseline LVEF was higher in iCMP patients but did not change in the follow-up echocardiogram. In patients with niCMP, LVEF increased from baseline to the follow-up echocardiogram, from 41.9 ± 15.6 to 44.8 ± 15.0 (*p* < 0.0001). Hospitalization and all causes of death were higher in iCMP patients. 

Table 2 shows the clinical characteristics and echocardiographic data for women and men with HF from nonischemic and ischemic CMP.

Women with niCMP were older and had a higher prevalence of HFpEF, diabetes, pacemakers and CRT implantations than men with niCMP. Women also had higher LVEF and lower LVDD in baseline and follow-up echocardiograms. In niCMP, the number of comorbidities, hospitalizations, and all causes of death were similar between women and men. Women with iCMP were older and had a higher prevalence of HFpEF. The number of comorbidities, coronary revascularization, pacemaker implantation, and hospitalizations were similar in women and men, but women had a lower incidence of all causes of death (24.5 vs. 29.8; *p* = 0.004). Men had a higher prevalence of myocardial infarction, diabetes, CKD, AF, and HFrEF. Men received more ICD and CRT implantations. Over a 3-year follow-up period, the cumulative incidence of death in iCMP was higher in men than in women (*p* < 0.001) but similar in niCMP (Figure 1).

The cumulative incidence of death in iCMP was higher in men than in women with HFrEF (*p* < 0.001) (Figure 2) but similar in iCMP with HFmrEF (Figure 3) and HFpEF (Figure 4). The cumulative incidence of death in niCMP was similar between women and men for all HF phenotypes (Figure 2, Figure 3 and Figure 4).

### Multivariate Analysis and Predictors of All Causes of Death

Table 3, Table 4 and Table 5 show the Cox proportional hazards ratios and the chi-square score of the Cox proportional hazards model of all causes of death. Cox regression was adjusted for confounders such as age, sex, MI, diabetes, stroke, CKD, AF, baseline LVEF, myocardial revascularization (percutaneous coronary intervention and coronary artery bypass graft), device implantation (pacemaker, internal cardiac defibrillator, and cardiac resynchronization therapy), ischemic, idiopathic, and hypertensive cardiomyopathies.

Table 3 shows the Cox regression analysis results for the main predictors of death in all patients and women and men for all HF etiologies. CKD, diabetes, stroke, age, lower baseline LVEF, MI, device implantation, and revascularization were the main death predictors for all patients, women and men. AF was a predictor of death for women and men.

Table 4 shows the Cox regression analysis results for the main predictors of death for niCMP in all patients and women, and men. CKD, diabetes, stroke, AF, age, lower baseline LVEF, device implantation, and idiopathic CMP were the main death predictors for all patients and men. CKD, diabetes, stroke, AF, age, and lower baseline LVEF were the main death predictors for women.

Table 5 shows the Cox regression analysis results for the main predictors of death in all patients and women and men for iCMP. CKD, stroke, diabetes, AF, MI, myocardial revascularization, age, and LVEF baseline in all patients, stroke, CKD, diabetes, AF, myocardial revascularization, MI, and age in women, and CKD, diabetes, stroke, AF, MI, age and LVEF baseline were the main predictors of death. Myocardial revascularization was an independent predictor of death only in women.

## 4. Discussion

This study demonstrates a lower incidence of death in women with HF due to ischemic CMP than in men, while no significant sex-based differences were observed in nonischemic CMP. Women presented with different clinical characteristics, including older age, higher LVEF, and lower LVDD, compared to men in both ischemic and nonischemic CMP. However, sex was not an independent variable associated with all-cause mortality in either subgroup.

Our results were similar to that observed in previous studies of HF from ischemic and idiopathic CMP [8,9,10]. The analysis of two recent studies in patients with HFrEF showed that women were older and had a higher prevalence of obesity and hypertension. In these studies, women also had fewer comorbidities except hypertension and a lesser risk of hospitalization [11].

The lower incidence of death observed in women with HF due to ischemic cardiomyopathy (iCMP) aligns with previous studies that reported better survival rates in women than men with HF [12,13]. These findings may be attributed to several factors, including hormonal differences, cardiac remodeling patterns, and therapy response. Estrogen, for instance, has been associated with cardioprotective effects, including favorable effects on endothelial function, vasodilation, and antioxidant activity, which may contribute to better outcomes in women [14,15]. However, it should be noted that the role of hormones in HF outcomes is complex and multifactorial, and further research is warranted to understand the underlying mechanisms involved in premenopausal and the partial persistence of beneficial effects in postmenopausal women.

Estrogen exerts a cardioprotective effect in HF by inhibiting sympathetic activity and the renin-angiotensin-aldosterone system, decreasing renin levels, angiotensin-converting enzyme activity, AT1 receptor density, aldosterone production, and increasing AT2 receptor density. Estrogen increases natriuretic peptides that intensify the renin-angiotensin-aldosterone system inhibition, promotes better endothelial response to injury, prevents left ventricular remodeling and diastolic function, and protects the coronary microvasculature [16,17]. Estrogen protection at the cellular level is primarily achieved by increasing anti-oxidative defenses and maintaining mitochondrial integrity [18]. These pathophysiological mechanisms of estrogens are probably partly responsible for the greater protection of myocardial cells observed in women. Estrogen protection at the cellular level is primarily achieved by increasing anti-oxidative defenses and maintaining mitochondrial integrity.

In addition to the lower mortality rates, women in both iCMP and niCMP presented with distinct clinical characteristics compared to men. Women were older and had higher LVEF and lower LVDD than men in both subgroups. These differences may affect disease progression, outcomes, and treatment response. Older age has been associated with worse outcomes in HF, and the higher LVEF and lower LVDD values observed in women may reflect differences in cardiac remodeling, contractile function, and prognosis [19,20].

The study findings also revealed an intriguing observation regarding the female and male populations with iCMP enrolled in the research. It was noted that the male participants exhibited a lower LVEF at baseline compared to their female counterparts. Table 2 provides further insights into this phenomenon, indicating that HFrEF occurred in 41.4% of men, while in women, this condition was observed in only 29.8% of cases. This discrepancy in HFrEF prevalence between genders highlights a potentially significant difference in the pathophysiological mechanisms underlying iCMP in men and women. Interestingly, the study found that baseline LVEF did not appear to impact the incidence of death in women with iCMP substantially. Despite the variations in LVEF, the mortality risk remained comparable among women with differing levels of LVEF at baseline. This intriguing result suggests that other factors beyond LVEF may be more dominant in determining the prognosis and outcomes for women with iCMP. One plausible explanation for this phenomenon could be attributed to the marked differences observed between the sexes regarding LVEF distribution. Notably, 53% of women with iCMP had preserved EF, whereas only 36% of men exhibited this preserved EF pattern. These patterns of LVEF distribution might influence the relationship between baseline LVEF and mortality risk in women with iCMP. 

However, despite the observed differences in clinical characteristics, sex was not found to be an independent variable associated with all-cause mortality in either iCMP or niCMP. These findings were similar to those previously observed in our chronic HF population [21] and suggest that other factors, such as atherosclerotic disease severity, comorbidities, genetic factors, or treatment strategies, may significantly influence HF outcomes. Likewise, HF severity, reflected by New York Heart Association (NYHA) functional class, biomarkers such as N-terminal pro-B-type natriuretic peptide [22] and echocardiographic parameters [23] is strongly associated with prognosis in HF. In our study, men had a higher prevalence of HFrEF than women, and previous studies showed a worse HFrEF prognosis than HFmrEF and HFpEF [13,24]. Comorbidities, including older age, hypertension, diabetes, CKD, stroke, and AF, can further complicate HF management and influence mortality risk [25]. The older age and the comparable number of comorbidities observed in women and men with niCMP and iCMP in our study may also explain why sex was not an independent variable for all causes of death. Genetic factors, such as specific gene polymorphisms or variations, may also contribute to individual variations in disease progression and response to treatment [26,27]. Nevertheless, genetic data were unavailable in our study. Treatment strategies can significantly impact HF outcomes, including optimal medical therapy, revascularization procedures, and device therapies [28]. The utilization and adherence to these therapies may vary between sexes, potentially influencing outcomes. However, in our study, women and men had the same treatment for almost all treatment strategies except for a higher ICD and CRT implantation in men with iCMP.

Future research should aim to identify and understand the interplay of these variables with sex-related factors to provide a more comprehensive understanding of HF outcomes. Large-scale prospective studies incorporating comprehensive clinical, genetic, and treatment data are needed to elucidate the complex interactions and identify potential targets for personalized interventions. Additionally, assessing long-term outcomes, including cardiovascular events, hospitalizations, and quality of life, would provide a more holistic view of the impact of sex and other factors on HF prognosis.

### Study Limitations

Our study has some limitations. It is a retrospective study in a specialized tertiary-care center where selection biases may occur, including patients with a more complex clinical picture. An adequate definition of symptoms is missing, especially the NYHA functional class of dyspnea and other variables associated with a worse prognosis, such as ventricular arrhythmia and a 6-min walk test. We were also unable to detail the cause of death adequately. Our analysis included cardiac and non-cardiac causes, including the deaths from COVID-19, which occurred between the pandemic months of March to September 2020. Finally, adequate information regarding drug treatment and dosages is also missing. However, our center advocates that the treatment of HF be as close as possible to current ‘Get With The Guidelines^®^’ care. Additionally, the study did not explore the potential impact of hormonal status, menopausal status, or hormone replacement therapy on HF outcomes, which could be relevant factors in sex-based differences.

## 5. Conclusions

In conclusion, this study has presented significant findings regarding the impact of ischemic cardiomyopathy (CMP) on women with heart failure (HF). Notably, it has revealed a decreased incidence of mortality in women with HF due to ischemic CMP compared to other etiologies. Furthermore, the research has shed light on notable distinctions in clinical characteristics between male and female patients. Nevertheless, while gender itself was not identified as an independent variable associated with all-cause mortality in either ischemic CMP or nonischemic CMP, the study underscores the influential role of other critical factors. These factors encompass the severity of HF disease, the presence of comorbidities such as coronary atherosclerotic severity, and the strategies employed for treatment. Recognizing these multifactorial influences is paramount in optimizing HF management and enhancing outcomes for both male and female patients. By acknowledging the diverse contributing elements, healthcare professionals can tailor treatment approaches to better suit individual needs and augment the quality of life for HF patients. This study highlights the importance of recognizing the unique characteristics of women with ischemic CMP and emphasizing the need for a holistic understanding of HF, considering the interconnected web of contributing factors. Through continued research and a commitment to applying the knowledge gained, we can strive to improve the lives of all individuals affected by heart failure, regardless of gender. Novel insights from studies like this are invaluable in propelling the medical community toward more relevant and effective strategies for managing HF and promoting better health outcomes for everyone.

## Figures and Tables

**Figure 1 jcm-12-05323-f001:**
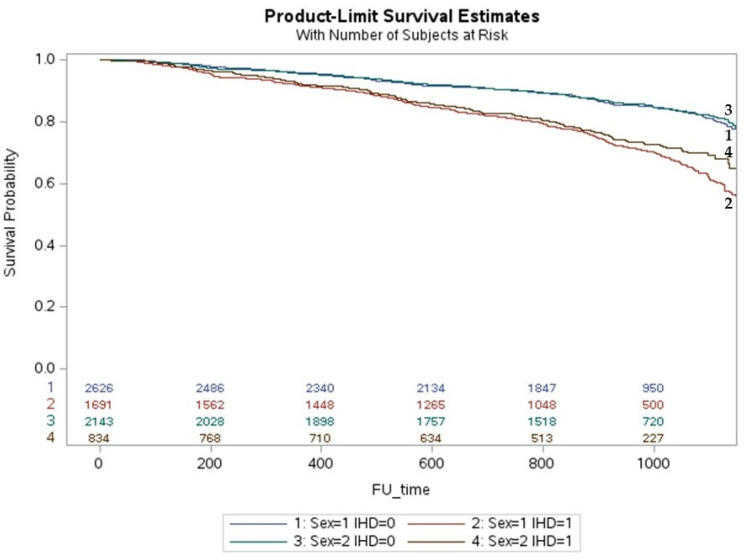
The life-table survival curves of women and men with chronic heart failure and nonischemic and ischemic cardiomyopathies. (sex = 1: men; sex = 2: women; IHD = 0: nonischemic cardiomyopathy; IHD = 1: ischemic cardiomyopathy).

**Figure 2 jcm-12-05323-f002:**
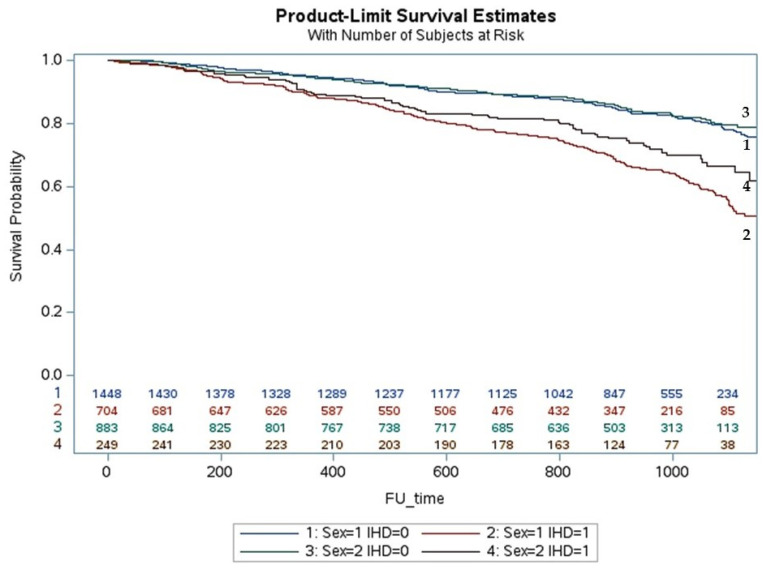
The life-table survival curves of women and men with chronic heart failure, reduced ejection fraction, and nonischemic and ischemic cardiomyopathies. (sex = 1: men; sex = 2: women; IHD = 0: nonischemic cardiomyopathy; IHD = 1: ischemic cardiomyopathy).

**Figure 3 jcm-12-05323-f003:**
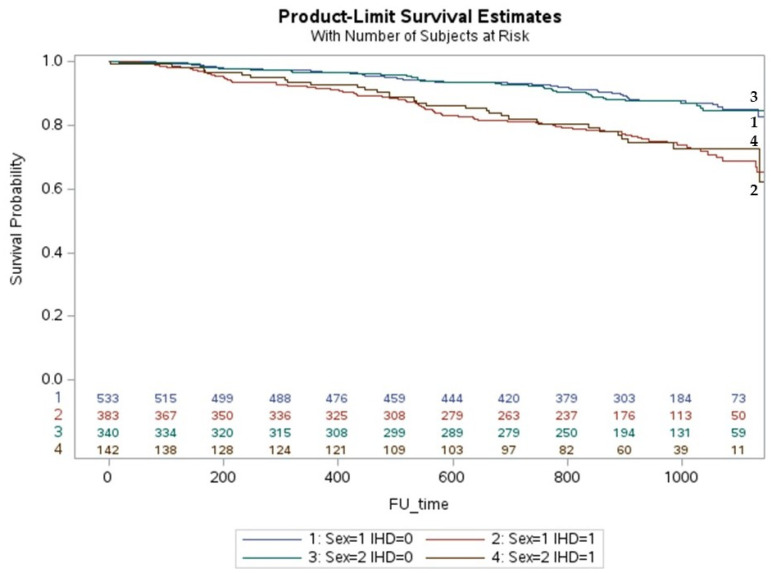
The life-table survival curves of women and men with chronic heart failure, mildly reduced ejection fraction, and nonischemic and ischemic cardiomyopathies. (sex = 1: men; sex = 2: women; IHD = 0: nonischemic cardiomyopathy; IHD = 1: ischemic cardiomyopathy).

**Figure 4 jcm-12-05323-f004:**
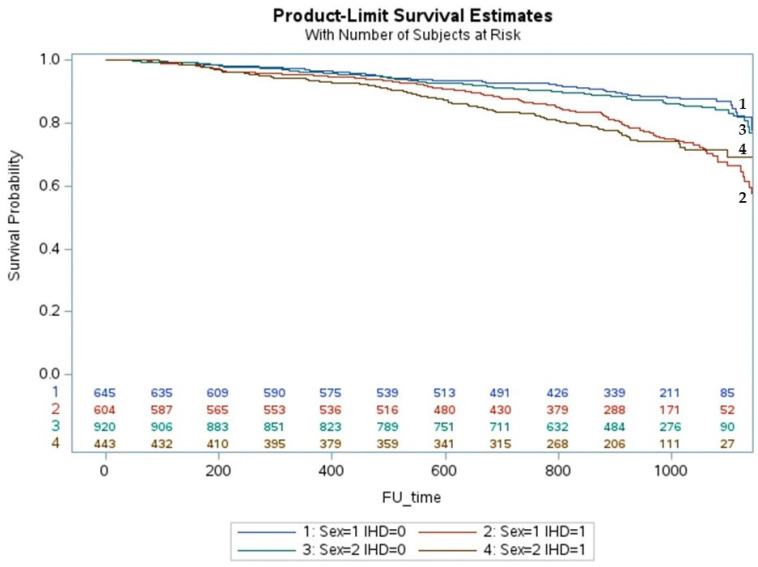
The life-table survival curves of women and men with chronic heart failure, preserved ejection fraction, and nonischemic and ischemic cardiomyopathies. (sex = 1: men; sex = 2: women; IHD = 0: nonischemic cardiomyopathy; IHD = 1: ischemic cardiomyopathy).

**Table 1 jcm-12-05323-t001:** Clinical characteristics and echocardiographic data in all patients with HF and those with nonischemic and ischemic CMP.

	All Patients	niCMP	iCMP	*p*
	*n* = 7483	*n* = 4883 (65.2)	*n* = 2600 (34.8)	
Age (Years)	64.26 ± 14.23	61.90 ± 14.97	68.71 ± 11.46	<0.001
Female (%)	3066 (41.0)	2205 (45.2)	861 (33.1)	<0.001
Comorbidities				
Myocardial infarction (%)	1350 (18.0)	3 (0.04)	1347 (51.8)	<0.001
Diabetes (%)	1496 (20.0)	630 (12.9)	866 (34.8)	<0.001
Chronic kidney disease (%)	817 (10.9)	401 (8.2)	416 (16.0)	<0.001
Stroke (%)	317 (4.2)	152 (3.1)	165 (6.4)	<0.001
Atrial fibrillation (%)	1356 (18.1)	897 (18.4)	459 (17.7)	0.4439
Number of comorbidities (%)	3659 (48.9)	1636 (33.5)	2023 (77.8)	<0.001
*n* = 1	2281 (30.5)	1209 (24.8)	1072 (41.2)	
*n* = 2	1009 (13.5)	348 (7.13)	661 (25.4)	
*n* = 3	300 (4.0)	72 (1.47)	228 (8.77)	
*n* ≥ 4	69 (0.9)	7 (0.14)	62 (2.38)	
Medication				
ACE inhibitor or BRA	4482 (59.9)	2940 (60.2)	1549 (59.6)	0.790
Beta-blocker	3839 (51.3)	2471 (50.6)	1344 (51.7)	0.610
Spironolactone	2095 (28.0)	1323 (27.1)	762 (29.3)	0.128
Diuretics	2499 (33.4)	2217 (45.4)	1199 (46.1)	0.719
Surgical intervention				
Coronary artery bypass graft (%)	741 (9.9)	0 (0.00)	741 (28.5)	<0.001
Percutaneous coronary intervention (%)	277 (3.7)	1 (0.02)	276 (10.6)	<0.001
Pacemaker implantation (%)	467 (6.2)	372 (7.6)	95 (3.7)	<0.001
Implantable cardioverter defibrillators (%)	224 (3.0)	116 (2.4)	108 (4.2)	<0.001
Cardiac resynchronization therapy (%)	275 (3.7)	221 (4.5)	54 (2.1)	<0.001
Transplant (%)	175 (2.3)	141 (2.9)	34 (1.3)	<0.001
Hospitalization (%)	2236 (30.0)	1279 (26.2)	957 (36.8)	<0.001
Echocardiogram				
LVEF baseline (%)	43.01 ± 15.39	41.87 ± 15.55	45.14 ± 14.86	<0.001
LVEF final (%)	44.87 ± 14.82	44.76 ± 14.99 ^a^	45.08 ± 14.47	0.364
LVDD baseline (mm)	58.22 ± 9.57	59.22 ± 9.88	56.31 ± 8.66	<0.001
LVDD final (mm)	57.72 ± 9.94	58.15 ± 10.31 ^b^	56.90 ± 9.15 ^b^	<0.001
Type of heart failure				
Reduced EF (%)	3359 (44.9)	2383 (48.8)	976 (37.5)	<0.001
Mildly reduced EF (%)	1436 (19.2)	896 (18.4)	540 (20.8)	0.0114
Preserved EF (%)	2688 (35.9)	1604 (32.9)	1084 (41.7)	<0.001
Death (%)	1475 (19.5)	745 (15.3)	730 (28.1)	<0.001

Values are mean ± SD or *n* (%). ^a^
*p* < 0.0001 (LVEF final vs. baseline in niCMP). ^b^
*p* < 0.0001 (LVDD final vs. baseline in niCMP and in iCMP). EF = ejection fraction; LVDD = left ventricular diastolic diameter; LVEF = left ventricular ejection fraction.

**Table 2 jcm-12-05323-t002:** Clinical characteristics and echocardiographic data in women and men with HF and nonischemic and ischemic CMP.

	niCMP *n* = 4883 (65.2)			iCMP *n* = 2600 (34.8)		
	Men *n* = 2678 (54.8)	Women *n* = 2205 (45.2)	*p*	Men *n* = 1736 (66.9)	Women *n* = 861 (33.1)	*p*
Age (Years)	60.18 ± 14.24	64.0 ± 15.5	<0.001	68.12 ± 11.05	69.9 ± 12.18	<0.001
Comorbidities						
Myocardial infarction (%)	0 (0.00)	3 (0.14)	0.056	921 (53.0)	426 (49.5)	0.094
Diabetes (%)	306 (11.4)	324 (14.7)	<0.001	561 (32.3)	305 (35.4)	0.107
Chronic kidney disease (%)	245 (9.2)	156 (7.1)	0.009	308 (17.7)	108 (12.5)	<0.001
Stroke (%)	94 (3.5)	58 (2.6)	0.078	111 (6.4)	54 (6.3)	0.913
Atrial fibrillation (%)	539 (20.1)	358 (16.2)	<0.001	328 (18.9)	131 (15.2)	0.022
Number of comorbidities (%)	916 (34.2)	720 (32.7)	0.334	1362 (78.5)	661 (76.8)	0.126
*n* = 1	669 (25.0)	540 (24.5)		705 (40.5)	367 (42.6)	
*n* = 2	203 (7.6)	145 (6.6)		449 (25.8)	212 (24.6)	
*n* = 3	42 (1.6)	30 (1.4)		159 (9.1)	69 (8.0)	
*n* ≥ 4	2 (0.1)	5 (0.2)		49 (2.3)	13 (1.5)	
Medication						
ACE inhibitor or ARB	1649 (61.6)	1291 (58.5)	0.283	1062 (61.2)	487 (56.6)	0.254
Beta-blocker	1344 (50.2)	1127 (51.1)	0.713	910 (52.4)	434 (50.4)	0.585
Spironolactone	744 (27.8)	579 (26.3)	0.366	519 (29.9)	243 (28.2)	0.514
Diuretics	1245 (46.5)	972 (44.1)	0.302	783 (45.1)	416 (48.3)	0.350
Surgical intervention						
Coronary artery bypass graft (%)	0 (0)	0 (0)		511 (29.4)	230 (26.7)	0.156
Percutaneous coronary intervention (%)	1 (0.04)	0 (0)		187 (10.6)	89 (10.3)	0.746
Pacemaker implantation (%)	162 (6.1)	210 (9.5)	<0.001	64 (3.7)	31 (3.6)	0.919
Implantable cardioverter defibrillators (%)	74 (2.8)	42 (1.9)	0.050	88 (5.1)	20 (2.3)	0.001
Cardiac resynchronization therapy (%)	103 (3.9)	118 (5.4)	0.012	45 (2.6)	9 (1.1)	0.009
Transplant (%)	90 (3.4)	51 (2.3)	0.030	24 (1.4)	10 (1.2)	0.644
Hospitalization (%)	676 (25.2)	603 (27.4)	0.096	654 (37.6)	303 (35.2)	0.229
Echocardiogram						
LVEF baseline (%)	39.0 ± 14.44	45.4 ± 16.12	<0.001	43.3 ± 14.48	48.8 ± 14.95	<0.001
LVEF final (%)	42.14 ± 14.64 ^a^	47.9 ± 14.8 ^a^	<0.001	43.3 ± 14.19	48.6 ± 14.38	<0.001
LVDD baseline (mm)	61.9 ± 9.67	56.0 ± 9.121	<0.001	57.8 ± 8.42	53.2 ± 8.30	<0.001
LVDD final (mm)	60.3 ± 10.46 ^b^	54.8 ± 9.51 ^b^	<0.001	58.3 ± 8.90 ^b^	53.3 ± 8.43 ^c^	<0.001
Type of heart failure						
Reduced EF (%)	1479 (55.2)	904 (41.0)	<0.001	719 (41.4)	257 (29.8)	<0.001
Mildly reduced EF (%)	543 (20.3)	353 (16.0)	<0.001	392 (22.5)	483 (17.2)	0.002
Preserved EF (%)	656 (24.5)	948 (43.0)	<0.001	628 (36.1)	456 (53.0)	<0.001
Death (%)	420 (15.7)	325 (14.7)	0.361	519 (29.8)	211 (24.5)	0.004

Values are mean ± SD or *n* (%). ^a^
*p* < 0.0001 (LVEF final vs. baseline in women and men with niCMP). ^b^
*p* < 0.0001 (LVDD final vs. baseline in men with niCMP and iCMP and men with iCMP). ^c^
*p* = 0.022 0001 (LVDD final vs. baseline in women with iCMP) ACE = angiotensin-converting enzyme; ARB = angiotensin receptor blocker; EF = ejection fraction; LVDD = left ventricular diastolic diameter; LVEF = left ventricular ejection fraction.

**Table 3 jcm-12-05323-t003:** Cox regression analysis for all causes of death and the chi-square score of death predictors in all patients with heart failure adjusted for age, gender, myocardial infarction, diabetes, stroke, chronic kidney disease, atrial fibrillation, baseline left ventricular ejection fraction, myocardial revascularization (percutaneous coronary intervention and coronary artery bypass graft), device implantation (pacemaker, internal cardiac defibrillator, and cardiac resynchronization therapy), ischemic, idiopathic, and hypertensive cardiomyopathies.

	Variable	Hazard Ratio	95% Confidence Limits		Variable	Score of Chi-Square Test	*p*
All patients	CKD	3.24	2.89	3.63	CKD	976.46	<0.001
	Stroke	2.62	2.25	3.05	Diabetes	251.47	<0.001
	Diabetes	2.22	1.98	2.48	Stroke	224.30	<0.001
	MI	1.42	1.25	1.61	Age	74.90	<0.001
	Device	1.31	1.11	1.54	LVEF baseline	73.43	<0.001
	Revascularization	1.19	1.03	1.38	MI	33.01	<0.001
	Idiopathic CMP	1.15	1.00	1.32	Device	8.69	0.003
	Age	1.02	1.02	1.03	Revascularization	4.49	0.034
	LVEF baseline	0.99	0.98	0.99	Idiopathic	3.88	0.049
Women	CKD	3.54	2.90	4.30	CKD	446.85	<0.001
	Stroke	3.07	2.37	3.99	Diabetes	119.93	<0.001
	Diabetes	2.57	2.15	3.07	Stroke	106.32	<0.001
	AF	2.02	1.66	2.45	AF	65.22	<0.001
	Revascularization	1.30	1.02	1.66	Age	39.31	<0.001
	MI	1.25	1.00	1.55	LVEF baseline	13.93	<0.001
	Age	1.03	1.02	1.03	Revascularization	8.10	0.004
	LVEF baseline	0.99	0.98	0.99	MI	3.92	0.048
Men	CKD	2.92	2.54	3.36	CKD	557.33	<0.001
	Stroke	2.24	1.85	2.71	Stroke	136.14	<0.001
	Diabetes	1.92	1.67	2.20	Diabetes	123.55	<0.001
	AF	1.91	1.67	2.20	AF	83.88	<0.001
	MI	1.44	1.24	1.66	LVEF baseline	52.02	<0.001
	Device	1.34	1.09	1.64	Age	43.16	<0.001
	Revascularization	1.26	1.06	1.50	MI	34.55	<0.001
	Age	1.02	1.01	1.02	Device	5.77	0.016
	LVEF baseline	0.98	0.97	0.98	Revascularization	4.87	0.027

AF: atrial fibrillation; CKD: chronic kidney disease; CMP: cardiomyopathy; Device: device implantation; LVEF: left ventricular ejection fraction; MI: myocardial infarction.

**Table 4 jcm-12-05323-t004:** Cox regression analysis for all causes of death and the chi-square score of death predictors in patients with heart failure from nonischemic cardiomyopathy adjusted for age, gender, myocardial infarction, diabetes, stroke, chronic kidney disease, atrial fibrillation, baseline left ventricular ejection fraction, myocardial revascularization (percutaneous coronary intervention and coronary artery bypass graft), device implantation (pacemaker, internal cardiac defibrillator, and cardiac resynchronization therapy), ischemic, idiopathic, and hypertensive cardiomyopathies.

	Variables	Hazard Ratio	95% Confidence Limits		Variables	Score of Chi-Square Test	*p*
All patients	CKD	3.59	3.03	4.24	CKD	608.58	<0.001
	Stroke	3.00	2.39	3.77	AF	149.38	<0.001
	Diabetes	2.66	2.25	3.14	Diabetes	136.74	<0.001
	AF	2.27	1.94	2.66	Stroke	93.62	<0.001
	Device	1.41	1.14	1.75	LVEF baseline	26.90	<0.001
	Idiopathic	1.28	1.09	1.52	Age	35.18	<0.001
	Age	1.02	1.01	1.02	Device	11.82	0.001
	LVEF baseline	0.98	0.98	0.99	Idiopathic	8.53	0.004
Women	CKD	3.82	2.94	4.96	CKD	301.64	<0.001
	Diabetes	2.89	2.29	3.64	Diabetes	86.44	<0.001
	Stroke	2.39	1.63	3.52	AF	49.24	<0.001
	AF	2.00	1.56	2.57	Stroke	20.37	<0.001
	Age	1.02	1.02	1.03	Age	18.78	<0.001
	LVEF baseline	0.99	0.98	0.99	LVEF baseline	13.44	<0.001
Men	CKD	3.75	3.01	4.68	CKD	325.64	<0.001
	Stroke	3.70	2.78	4.92	Stroke	116.60	<0.001
	AF	2.60	2.12	3.19	AF	95.47	<0.001
	Diabetes	2.31	1.83	2.93	Diabetes	37.23	<0.001
	Device	1.52	1.13	2.05	LVEF baseline	29.34	<0.001
	Idiopathic	1.50	1.18	1.90	Device	14.15	<0.001
	Age	1.01	1.00	1.02	Idiopathic	8.20	0.004
	LVEF baseline	0.98	0.97	0.99	Age	8.63	0.003

AF: atrial fibrillation; CKD: chronic kidney disease; Device: device implantation; LVEF: left ventricular ejection fraction; MI: myocardial infarction.

**Table 5 jcm-12-05323-t005:** Cox regression analysis for all causes of death and the chi-square score of death predictors in patients with heart failure from ischemic cardiomyopathy adjusted for age, gender, myocardial infarction, diabetes, stroke, chronic kidney disease, atrial fibrillation, baseline left ventricular ejection fraction, myocardial revascularization (percutaneous coronary intervention and coronary artery bypass graft), device implantation (pacemaker, internal cardiac defibrillator, and cardiac resynchronization therapy), ischemic, idiopathic, and hypertensive cardiomyopathies.

	Variables	Hazard Ratio	95% Confidence Limits		Variables	Score of Chi-Square Test	*p*
All patients	CKD	2.53	2.16	2.95	CKD	286.12	<0.001
	Stroke	2.09	1.71	2.57	Stroke	85.80	<0.001
	Diabetes	1.93	1.66	2.24	Diabetes	73.63	<0.001
	AF	1.58	1.34	1.86	AF	35.05	<0.001
	MI	1.45	1.24	1.68	LVEF baseline	24.25	<0.001
	Revascularization	1.18	1.01	1.37	Age	39.57	<0.001
	Age	1.02	1.02	1.03	MI	22.10	<0.001
	LVEF baseline	0.98	0.98	0.99	Revascularization	4.61	0.032
Women	Stroke	3.91	2.72	5.62	CKD	116.26	<0.001
	CKD	3.39	2.51	4.58	Stroke	61.25	<0.001
	Diabetes	2.27	1.71	3.01	Diabetes	27.92	<0.001
	AF	1.93	1.40	2.66	AF	26.06	<0.001
	Revascularization	1.38	1.05	1.83	Age	16.72	<0.001
	MI	1.35	1.01	1.80	Revascularization	4.58	0.032
	Age	1.03	1.01	1.04	MI	4.26	0.039
Men	CKD	2.41	2.00	2.89	CKD	175.89	<0.001
	Diabetes	1.87	1.56	2.23	Diabetes	45.81	<0.001
	Stroke	1.80	1.40	2.30	Stroke	40.28	<0.001
	AF	1.48	1.22	1.80	LVEF baseline	29.06	<0.001
	MI	1.45	1.21	1.74	Age	26.61	<0.001
	Age	1.02	1.01	1.03	MI	18.16	<0.001
	LVEF baseline	0.98	0.97	0.99	AF	15.71	<0.001

AF: atrial fibrillation; CKD: chronic kidney disease; LVEF: left ventricular ejection fraction; MI: myocardial infarction.

## Data Availability

Data are unavailable due to privacy reasons.

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
