# Peer review of "Sex Differences in Prognosis of Heart Failure Due to Ischemic and Nonischemic Cardiomyopathy"

_jcm, 2023, doi:10.3390/jcm12165323_

Round 1

Reviewer 1 Report

The influence of gender differences in prognosis of heart failure has been previously investigated, by the same research group as well, among others. The original side of the present study would have been an analysis within two etiologic categories: de ischemic and non-ischemic cardiomyopathy.

Comments:

Tables or histograms with the statistics data of death predictors, with an accent on cardiovascular death, in men compared to women in the iCMP group, and the same for the niCMP group should be provided. A more clear scaling of the results, in terms of novelty and relevance for the management and treatment of HF is required.

Line 156: ‘Women with niCMP were older and had a higher prevalence of HFpEF, diabetes, pacemaker, and CRT implantations.’ – please specify in the text to whom you compare them: to women iCMP or to men niCMP?

Minor editing of English language are required, e.g.: line 234 - 'Our results were similar to that observed in previous studies'

Author Response

Reviewer 1.

Thanks for your time in reviewing our manuscript. The English language was edited. Below are the responses to the reviewer's suggestions.

The influence of gender differences in prognosis of heart failure has been previously investigated, by the same research group as well, among others. The original side of the present study would have been an analysis within two etiologic categories: de ischemic and non-ischemic cardiomyopathy.

Comments:

Tables or histograms with the statistics data of death predictors, with an accent on cardiovascular death, in men compared to women in the iCMP group, and the same for the niCMP group should be provided. A more clear scaling of the results, in terms of novelty and relevance for the management and treatment of HF is required.

RE: Unfortunately, we have only all causes of death. As stated in the study limitations, we could not accurately detail the cause of death. The novelty and relevance are detailed in the Conclusion section.

 Line 156: ‘Women with niCMP were older and had a higher prevalence of HFpEF, diabetes, pacemaker, and CRT implantations.’ – please specify in the text to whom you compare them: to women iCMP or to men niCMP?

RE: OK. Done.

Comments on the Quality of English Language

Minor editing of English language are required, e.g.: line 234 - 'Our results were similar to that observed in previous studies'

RE: OK. English language revision was done.

Reviewer 2 Report

Review Manuscript JCM # 2498940

The retrospective study“ Sex differences in prognosis of heart failure due to ischemic and nonischemic cardiomyopathy” was designed to determine sex-specific differences in death predictors of HF in patients with ischemic (iCMP) and nonischemic (niCMP) cardiomyopathy the effect of low carbohydrate diet (LC) in patients with diabetic cardiomyopathy (DMCM). The paper is well-written and discussion was based on sufficient literature review. This is an interesting study provided in large cohort (7487 patients) with over 2 year follow-up period. Nevertheless, the results of the study do not provide innovative information, but basically confirm the current state of knowledge about the prognosis and risk factors for death in ischemic cardiomyopathy. Namely authors observed that men with iCMP had higher incidence of death. The main predictors of death for iCMP in men and women were chronic kidney disease, diabetes, stroke, atrial fibrillation, age, and myocardial infarction.

Several issues raise objections and raise questions which, as a reviewer, I would like to address to the authors of the study.

1.      Where does the definition of ischemic cardiomyopathy come from? References are essential.  Lines 75-76.

The definition adopted by the authors of the study does not take into account the clinical situation, e.g. with 60% stenosis of the left main coronary artery and, for example, concomitant stenosis of other vessels (50-60%). On the other hand, it is known that 70% stenosis of the coronary artery in many situations may not cause symptoms in the form of contractility abnormalities, i.e. the effect of artery stenosis on the function of the myocardium.

I consider this to be a significant limitation of the study.

2.      Line 74 and 75 heart failure was diagnoses basing on Framingham criteria for HF and echocardiographic measurements. The only echocardiographic criterium according to guidelines used in the study was EF. But preserved EF was detected in 41.7% patients in iCMP and in 32,9% patients in niCMP. How HF was defined in those patients with preserved EF?

3.      Did you exclude patients with HCM or DCM cardiomyopathy from niCMP?

4.      Table 1 shows low incident of myocardial infarction in iCMP population only 18% (table 1) - I think there is a mistake as ischemic cardiomyopathy is the most often  diagnosed after acute coronary incident.

5.      Table 1. LVEF basal and final – this should be explained in the table legend and in main text (presumably lines 80-81)

6.      The authors use EF basal and EF baseline interchangeably. It needs to be unified.

7.      Table 3. What does LVEF1 mean?

8.      Why do the authors consider the presence of an implantable devices as a risk factor for death? (Table 3, 4) It suggests that the implantation of cardiac device itself increases the risk for death, what is wrong suggestion or conclusion. As the truth is that the presence of an ICD or CRT/CRT-D already indicates an increased risk of death, because only such patients (after cardiac arrest or with low EF) are implanted with the above devices.

9.      The iCMP male population enrolled in the study had a lower EF at baseline. According to Table 2, HFrEF occurred in 41.4% of men and only 29.8% in women. And surprisingly baseline LVEF did-not affect incidence of death in women in iCMP – this is worth discussing (maybe it resulted from differences between EF (53% women and only 36 % of men  in iCMP  had preserved EF).

10.  I suggest that the study should include one more analysis in subgroups that would possibly improve the research value of the study. Namely an analysis of mortality and risk factors for death in men and women with HFrEF, EFmrEF, HFpEF. Such an analysis could lead to interesting conclusions.

11.  Discussion: First of all, it is worth mentioning what is new in the results of this study.

Author Response

Thanks for your time in reviewing our manuscript. The English language was edited. Below are the responses to the reviewer's suggestions.

The retrospective study“ Sex differences in prognosis of heart failure due to ischemic and nonischemic cardiomyopathy” was designed to determine sex-specific differences in death predictors of HF in patients with ischemic (iCMP) and nonischemic (niCMP) cardiomyopathy the effect of low carbohydrate diet (LC) in patients with diabetic cardiomyopathy (DMCM). The paper is well-written and discussion was based on sufficient literature review. This is an interesting study provided in large cohort (7487 patients) with over 2 year follow-up period. Nevertheless, the results of the study do not provide innovative information, but basically confirm the current state of knowledge about the prognosis and risk factors for death in ischemic cardiomyopathy. Namely authors observed that men with iCMP had higher incidence of death. The main predictors of death for iCMP in men and women were chronic kidney disease, diabetes, stroke, atrial fibrillation, age, and myocardial infarction.

 Several issues raise objections and raise questions which, as a reviewer, I would like to address to the authors of the study.

  1. Where does the definition of ischemic cardiomyopathy come from? References are essential. Lines 75-76.

The definition adopted by the authors of the study does not take into account the clinical situation, e.g. with 60% stenosis of the left main coronary artery and, for example, concomitant stenosis of other vessels (50-60%). On the other hand, it is known that 70% stenosis of the coronary artery in many situations may not cause symptoms in the form of contractility abnormalities, i.e. the effect of artery stenosis on the function of the myocardium.

I consider this to be a significant limitation of the study.

RE: The definition of iCMP used in our study came from the study of Felker et al. (Felker GM, Shaw LK, O'Connor CM. A standardized definition of ischemic cardiomyopathy for use in clinical research. J Am Coll Cardiol. 2002 Jan 16;39(2):210-8. doi: 10.1016/s0735-1097(01)01738-7.) We only adjusted the 75% CAD stenosis in the Felker study to 70% used in our study. This was because estimating between 70% and 75% by qualitative CAD stenosis evaluation is challenging. The reference was included in the text.

You are right in relation to the discrete-moderate coronary lesions being responsible for iCMP, but if this occurred in our population, the number was very low, not compromising the study results.

You are also right in relation to asymptomatic patients with more than 70% coronary stenosis. The symptom absence in these patients probably occurs by known and unknown compensatory mechanisms. Nevertheless, these patients are still classified as having significant coronary disease.

  1. Line 74 and 75 heart failure was diagnoses basing on Framingham criteria for HF and echocardiographic measurements. The only echocardiographic criterium according to guidelines used in the study was EF. But preserved EF was detected in 41.7% patients in iCMP and 32,9% patients in niCMP. How HF was defined in those patients with preserved EF?

RE: Thanks for the observation of the preserved HF missing definition. The criteria were included in the Methods section.

  1. Did you exclude patients with HCM or DCM cardiomyopathy from niCMP?

RE: Yes. Patients with hypertrophic CMP were excluded.

  1. Table 1 shows low incident of myocardial infarction in iCMP population only 18% (table 1) - I think there is a mistake as ischemic cardiomyopathy is the most often diagnosed after acute coronary incident.

RE: Thanks again for the observation. The right percentage was 51.8% (1347/2600). The number in the Table was corrected.

  1. Table 1. LVEF basal and final – this should be explained in the table legend and in main text (presumably lines 80-81)

RE: Done. The definition was included in the main text.

  1. The authors use EF basal and EF baseline interchangeably. It needs to be unified.

RE: Done. Baseline instead basal.

  1. Table 3. What does LVEF1 mean?

RE: Means baseline LVEF. Corrected in the text.

  1. Why do the authors consider the presence of an implantable devices as a risk factor for death? (Table 3, 4) It suggests that the implantation of cardiac device itself increases the risk for death, what is wrong suggestion or conclusion. As the truth is that the presence of an ICD or CRT/CRT-D already indicates an increased risk of death, because only such patients (after cardiac arrest or with low EF) are implanted with the above devices.

RE: We do not consider cardiac device implantation as a risk factor but as a treatment intervention, and we checked if they were done more in women or men. Nevertheless, device implantation was included in the multivariable analysis because p<0.1 and was an independent variable for all causes of death. As you pointed out, these patients are probably already at high risk of death.

  1. The iCMP male population enrolled in the study had a lower EF at baseline. According to Table 2, HFrEF occurred in 41.4% of men and only 29.8% in women. And surprisingly baseline LVEF did-not affect incidence of death in women in iCMP – this is worth discussing (maybe it resulted from differences between EF (53% women and only 36 % of men in iCMP  had preserved EF).

RE: Done.

  1. I suggest that the study should include one more analysis in subgroups that would possibly improve the research value of the study. Namely an analysis of mortality and risk factors for death in men and women with HFrEF, EFmrEF, HFpEF. Such an analysis could lead to interesting conclusions.

RE: We analyzed the cumulative death rate using KM method for each HF phenotype. Three Figures were also included in the Results section.

  1. Discussion: First of all, it is worth mentioning what is new in the results of this study.

RE: Done.

Round 2

Reviewer 1 Report

The article was sufficiently improved for being published in JCM

Author Response

Dear Reviewer, thanks again for your time in reviewing my manuscript. 

Reviewer 2 Report

ROUND 2

REVIEWER: Thank you for answering my questions. Below I send further comments on the text of the manuscript, and in fact on the methodology used.

  1. Where does the definition of ischemic cardiomyopathy come from? References are essential. Lines 75-76.

The definition adopted by the authors of the study does not take into account the clinical situation, e.g. with 60% stenosis of the left main coronary artery and, for example, concomitant stenosis of other vessels (50-60%). On the other hand, it is known that 70% stenosis of the coronary artery in many situations may not cause symptoms in the form of contractility abnormalities, i.e. the effect of artery stenosis on the function of the myocardium.

I consider this to be a significant limitation of the study.

RE: The definition of iCMP used in our study came from the study of Felker et al. (Felker GM, Shaw LK, O'Connor CM. A standardized definition of ischemic cardiomyopathy for use in clinical research. J Am Coll Cardiol. 2002 Jan 16;39(2):210-8. doi: 10.1016/s0735-1097(01)01738-7.) We only adjusted the 75% CAD stenosis in the Felker study to 70% used in our study. This was because estimating between 70% and 75% by qualitative CAD stenosis evaluation is challenging. The reference was included in the text.

You are right in relation to the discrete-moderate coronary lesions being responsible for iCMP, but if this occurred in our population, the number was very low, not compromising the study results.

You are also right in relation to asymptomatic patients with more than 70% coronary stenosis. The symptom absence in these patients probably occurs by known and unknown compensatory mechanisms. Nevertheless, these patients are still classified as having significant coronary disease.

RE REVIEWER:

Felker and al. proposed definition of iCMP differing from definition adopted by the authors of present manuscript.

·         Patients with history of MI or revascularization (CABG or PCI)

·         Patients with 75% stenosis of left main or proximal LAD.

·         Patients with 75% stenosis of two or more epicardial vessels

In your study iCMP was diagnosed when there was more than 70% luminal reduction in at least one coronary artery with a significant area of myocardium.

Moreover, in the study by Felner et al. patients with HF symptoms and EF≤ 40% were analyzed, whereas in the reviewed manuscript preserved EF was detected in 41.7% 

In the population with preserved EF, it is hard to say if HF results from CAD, or arterial hypertension or diabetes or arrhythmia (e.g. AF). 70% stenosis in coronary artery, even when it is significant in functional tests is not equal with the diagnosis of ischemic cardiomyopathy.

Therefore, I still consider this to be a serious methodological flaw in the design of the study and the use of the term “cardiomyopathy”. In view of the above, I believe that patients with  preserved ejection fraction should be removed from the analysis. Analysis should be provided only for patients with HFmrEF and HFrEF in context of "cardiomyopathy". The observation of the population with HFmrEF could serve as some refinement of the analysis by Felker et al.

2.     Line 74 and 75 heart failure was diagnoses basing on Framingham criteria for HF and echocardiographic measurements. The only echocardiographic criterium according to guidelines used in the study was EF. But preserved EF was detected in 41.7% patients in iCMP and 32,9% patients in niCMP. How HF was defined in those patients with preserved EF?

RE: Thanks for the observation of the preserved HF missing definition. The criteria were included in the Methods section.

3.     Did you exclude patients with HCM or DCM cardiomyopathy from niCMP?

RE: Yes. Patients with hypertrophic CMP were excluded.

RE REVIEWER: and what about DCM patients in miCMP?

4.     Table 1 shows low incident of myocardial infarction in iCMP population only 18% (table 1) - I think there is a mistake as ischemic cardiomyopathy is the most often diagnosed after acute coronary incident.

RE: Thanks again for the observation. The right percentage was 51.8% (1347/2600). The number in the Table was corrected.

5.     Table 1. LVEF basal and final – this should be explained in the table legend and in main text (presumably lines 80-81)

RE: Done. The definition was included in the main text.

6.     The authors use EF basal and EF baseline interchangeably. It needs to be unified.

RE: Done. Baseline instead basal.

7.     Table 3. What does LVEF1 mean?

RE: Means baseline LVEF. Corrected in the text.

8.     Why do the authors consider the presence of an implantable devices as a risk factor for death? (Table 3, 4) It suggests that the implantation of cardiac device itself increases the risk for death, what is wrong suggestion or conclusion. As the truth is that the presence of an ICD or CRT/CRT-D already indicates an increased risk of death, because only such patients (after cardiac arrest or with low EF) are implanted with the above devices.

RE: We do not consider cardiac device implantation as a risk factor but as a treatment intervention, and we checked if they were done more in women or men. Nevertheless, device implantation was included in the multivariable analysis because p<0.1 and was an independent variable for all causes of death. As you pointed out, these patients are probably already at high risk of death.

RE REVIEWER: the presence of devices is related with low EF or cardiac arrest episode. So it will give always you statistical significance but not related with device implantation itself but with all criteria’s that we use for device implantation, that are already present in Cox regression analysis. Moreover, if we were to analyze the mortality of patients with a low fraction with and without implanted devices, it would turn out that people with implanted devices have a lower mortality rate. And this is already evident in present Cox's analysis. For this reason, according to the guidelines, these devices are used as a form of therapy. Therefore, the devices shoul be removed from the mortality risk analysis, because including this point in the statistical analysis is a methodological error. As mentioned in the previous review the presence of device is not a risk factor as it is shown in the present study.

  1. The iCMP male population enrolled in the study had a lower EF at baseline. According to Table 2, HFrEF occurred in 41.4% of men and only 29.8% in women. And surprisingly baseline LVEF did-not affect incidence of death in women in iCMP – this is worth discussing (maybe it resulted from differences between EF (53% women and only 36 % of men in iCMP had preserved EF).

RE: Done.

  1. I suggest that the study should include one more analysis in subgroups that would possibly improve the research value of the study. Namely an analysis of mortality and risk factors for death in men and women with HFrEF, EFmrEF, HFpEF. Such an analysis could lead to interesting conclusions.

RE: We analyzed the cumulative death rate using KM method for each HF phenotype. Three Figures were also included in the Results section.

  1. Discussion: First of all, it is worth mentioning what is new in the results of this study.

RE: Done.

Author Response

Dear Reviewer, thanks again for your time in reviewing my manuscript. Below are the answers to your questions highlighted in red.

REVIEWER: Thank you for answering my questions. Below I send further comments on the text of the manuscript, and in fact on the methodology used.

Where does the definition of ischemic cardiomyopathy come from? References are essential. Lines 75-76.

The definition adopted by the authors of the study does not take into account the clinical situation, e.g. with 60% stenosis of the left main coronary artery and, for example, concomitant stenosis of other vessels (50-60%). On the other hand, it is known that 70% stenosis of the coronary artery in many situations may not cause symptoms in the form of contractility abnormalities, i.e. the effect of artery stenosis on the function of the myocardium.

I consider this to be a significant limitation of the study.

RE: The definition of iCMP used in our study came from the study of Felker et al. (Felker GM, Shaw LK, O'Connor CM. A standardized definition of ischemic cardiomyopathy for use in clinical research. J Am Coll Cardiol. 2002 Jan 16;39(2):210-8. doi: 10.1016/s0735-1097(01)01738-7.) We only adjusted the 75% CAD stenosis in the Felker study to 70% used in our study. This was because estimating between 70% and 75% by qualitative CAD stenosis evaluation is challenging. The reference was included in the text.

You are right in relation to the discrete-moderate coronary lesions being responsible for iCMP, but if this occurred in our population, the number was very low, not compromising the study results.

You are also right in relation to asymptomatic patients with more than 70% coronary stenosis. The symptom absence in these patients probably occurs by known and unknown compensatory mechanisms. Nevertheless, these patients are still classified as having significant coronary disease.

RE REVIEWER:

Felker and al. proposed definition of iCMP differing from definition adopted by the authors of present manuscript.

  • Patients with history of MI or revascularization (CABG or PCI)

  • Patients with 75% stenosis of left main or proximal LAD.

  • Patients with 75% stenosis of two or more epicardial vessels

In your study iCMP was diagnosed when there was more than 70% luminal reduction in at least one coronary artery with a significant area of myocardium.

Moreover, in the study by Felner et al. patients with HF symptoms and EF≤ 40% were analyzed, whereas in the reviewed manuscript preserved EF was detected in 41.7%

In the population with preserved EF, it is hard to say if HF results from CAD, or arterial hypertension or diabetes or arrhythmia (e.g. AF). 70% stenosis in coronary artery, even when it is significant in functional tests is not equal with the diagnosis of ischemic cardiomyopathy.

Therefore, I still consider this to be a serious methodological flaw in the design of the study and the use of the term “cardiomyopathy”. In view of the above, I believe that patients with  preserved ejection fraction should be removed from the analysis. Analysis should be provided only for patients with HFmrEF and HFrEF in context of "cardiomyopathy". The observation of the population with HFmrEF could serve as some refinement of the analysis by Felker et al.

 RE RE: OK. Done

  1. Line 74 and 75 heart failure was diagnoses basing on Framingham criteria for HF and echocardiographic measurements. The only echocardiographic criterium according to guidelines used in the study was EF. But preserved EF was detected in 41.7% patients in iCMP and 32,9% patients in niCMP. How HF was defined in those patients with preserved EF?

RE: Thanks for the observation of the preserved HF missing definition. The criteria were included in the Methods section.

  1. Did you exclude patients with HCM or DCM cardiomyopathy from niCMP?

RE: Yes. Patients with hypertrophic CMP were excluded.

RE REVIEWER: and what about DCM patients in miCMP?

RE RE: Yes, it was included in niCMP.

  1. Table 1 shows low incident of myocardial infarction in iCMP population only 18% (table 1) - I think there is a mistake as ischemic cardiomyopathy is the most often diagnosed after acute coronary incident.

RE: Thanks again for the observation. The right percentage was 51.8% (1347/2600). The number in the Table was corrected.

  1. Table 1. LVEF basal and final – this should be explained in the table legend and in main text (presumably lines 80-81)

RE: Done. The definition was included in the main text.

  1. The authors use EF basal and EF baseline interchangeably. It needs to be unified.

RE: Done. Baseline instead basal.

  1. Table 3. What does LVEF1 mean?

RE: Means baseline LVEF. Corrected in the text.

  1. Why do the authors consider the presence of an implantable devices as a risk factor for death? (Table 3, 4) It suggests that the implantation of cardiac device itself increases the risk for death, what is wrong suggestion or conclusion. As the truth is that the presence of an ICD or CRT/CRT-D already indicates an increased risk of death, because only such patients (after cardiac arrest or with low EF) are implanted with the above devices.

RE: We do not consider cardiac device implantation as a risk factor but as a treatment intervention, and we checked if they were done more in women or men. Nevertheless, device implantation was included in the multivariable analysis because p<0.1 and was an independent variable for all causes of death. As you pointed out, these patients are probably already at high risk of death.

RE REVIEWER: the presence of devices is related with low EF or cardiac arrest episode. So it will give always you statistical significance but not related with device implantation itself but with all criteria’s that we use for device implantation, that are already present in Cox regression analysis. Moreover, if we were to analyze the mortality of patients with a low fraction with and without implanted devices, it would turn out that people with implanted devices have a lower mortality rate. And this is already evident in present Cox's analysis. For this reason, according to the guidelines, these devices are used as a form of therapy. Therefore, the devices shoul be removed from the mortality risk analysis, because including this point in the statistical analysis is a methodological error. As mentioned in the previous review the presence of device is not a risk factor as it is shown in the present study.

RE RE: Thank you for your comments, but I respectfully disagree with your comments regarding the implantation of devices as a variable to be excluded from the Cox analysis. As a variable with p<0.1 in the univariate analysis, it should be included in the model in the same way surgical interventions were included. Both variables, devices and surgeries, were interventions applied in the niCMP and iCMP groups in the various CHF phenotypes, not only in low EF or cardiac arrest episodes.

The iCMP male population enrolled in the study had a lower EF at baseline. According to Table 2, HFrEF occurred in 41.4% of men and only 29.8% in women. And surprisingly baseline LVEF did-not affect incidence of death in women in iCMP – this is worth discussing (maybe it resulted from differences between EF (53% women and only 36 % of men in iCMP had preserved EF).

RE: Done.

I suggest that the study should include one more analysis in subgroups that would possibly improve the research value of the study. Namely an analysis of mortality and risk factors for death in men and women with HFrEF, EFmrEF, HFpEF. Such an analysis could lead to interesting conclusions.

RE: We analyzed the cumulative death rate using KM method for each HF phenotype. Three Figures were also included in the Results section.

Discussion: First of all, it is worth mentioning what is new in the results of this study.

RE: Done.